Overexpression of the LcCUC2-like gene in Arabidopsis thaliana alters the cotyledon morphology and increases rosette leaf number

Wen Shaoying 1
Li Jiayu 1
Hao Ziyuan 1
Wei Lingmin 1
Ma Jikai 1
Zong Yaxian 1
Li Huogen hgli@njfu.edu.cn 1
Key Laboratory of Forest Genetics & Biotechnology of Ministry of Education, Co-Innovation Center for Sustainable Forestry in Southern China, Nanjing Forestry University , Nanjing , Jiangsu , China
Bonetta Dario
Electronic publication date: 2022 Feb 2
Publication date: 2022
Volume: 10
Electronic Location ID: e12615
Received 2021 Feb 26; Accepted 2021 Nov 18
Copyright: ©2022 Wen et al.
Copyright year: 2022
Copyright holder: Wen et al.
License: This is an open access article distributed under the terms of the Creative Commons Attribution License, which permits unrestricted use, distribution, reproduction and adaptation in any medium and for any purpose provided that it is properly attributed. For attribution, the original author(s), title, publication source (PeerJ) and either DOI or URL of the article must be cited.
License URL: https://creativecommons.org/licenses/by/4.0/

Keywords: Liriodendron Chinense, Leaf development, Cotyledon, Rosette leaf, LcCUC2-like

Funding: The National Natural Science Foundation of China 31770718 31470660 The Priority Academic Program Development of Jiangsu Higher Education Institutions (PAPD) This work was supported by the National Natural Science Foundation of China (31770718, 31470660) and the Priority Academic Program Development of Jiangsu Higher Education Institutions (PAPD). The funders had no role in study design, data collection and analysis, decision to publish, or preparation of the manuscript.

==============================
Background

The unique ‘mandarin jacket’ leaf shape is the most famous trait of Liriodendron chinense and this characteristic gives L. chinense aesthetic and landscaping value. However, the underlying regulatory mechanism of genes involved in the leaf development of L. chinense has remained unclear.

Methods

Based on transcriptome data of leaves at different developmental stages from L. chinense, we identified differentially expression genes (DEGs) functioning in leaf development. A candidate gene named LcCUC2-like (LcCUC2L) had high similarity in sequence with Arabidopsis thaliana CUC2, and used for further research. We isolated the full-length LcCUC2L gene and its promoter from L. chinense. Subsequently, we analyzed the function of the LcCUC2L gene and its promoter activity via transformation into A. thaliana.

Results

In this study, we found that the LcCUC2L and AtCUC2 are homologous in sequence but not homologous in function. Unlike the role of AtCUC2 in leaf serration and SAM formation, the LcCUC2L mainly regulates cotyledon development and rosette leaf number. Histochemical β-glucuronidase (GUS) staining revealed that LcCUC2L was expressed in the cotyledons of A. thaliana seedlings, indicating that the LcCUC2L may play a role in cotyledon development. Ectopic expression of LcCUC2L resulted in long, narrow cotyledons without petioles, abnormal lamina epidermis cells and defective vascular tissue in cotyledons, and these results were consistent with the LcCUC2L expression pattern. Further analysis showed that overexpression of LcCUC2L also induced numerous rosette leaves. Also, LcCUC2L and other related genes showed a severe response in L. chinense by introducing exogenous auxin stimulation, partly revealed that LcCUC2L affects the leaf development by regulating the auxin content.

Conclusions

These results suggest that LcCUC2L may play a critical role in leaf development and morphogenesis in L. chinense, and our findings provide insight into the molecular mechanisms of leaf development in L. chinense.

Introduction

The leaf, as an indispensable plant organ, functions in photosynthesis, respiration, and photoperception (Li et al., 2007). Leaf shape is one of the most important characteristics affecting plant productivity and survival. Previous studies have shown that compared with unlobed leaves, lobed leaves have larger specific leaf areas, which can enhance the competitive ability of the plant for light resources and improve photosynthetic rates (Semchenko & Zobel, 2007). Moreover, dissected leaves allow adjustment of leaf surface temperature and allow strong adaptability to stress (Vogel, 2009). In recent years, researchers have paid much attention to the regulatory mechanisms of leaf shape development (Bilsborough et al., 2011). In addition to environmental factors, many genetic factors, such as complex gene interactions, gene expression patterns, and microRNAs, and active hormonal regulation influence leaf shape development (Dkhar & Pareek, 2014). To date, many studies of leaf shape development have focused on herbaceous plants such as A. thaliana (Kessler & Sinha, 2004), Cardamine hirsuta (Barkoulas et al., 2008), Medicago truncatula (Peng et al., 2011), and Solanum lycopersicum (Chitwood et al., 2014). However, in woody plants, the mechanisms of leaf shape development remain poorly understood.

Liriodendron chinense (Hemsl.) Sarg, a valuable endemic tree belonging to the magnolia family (Magnoliaceae), is distributed mainly in southern China and northern Vietnam (Fang, 1994; Wang, 1997). L. chinense is an attractive species for ornamental use, with a distinctive leaf shape similar to that of the traditional Chinese mandarin jacket. Normally, L. chinense has one deep lobe on each side of the leaf margin (Yang et al., 2014). Due to its unique leaf shape, some investigators have focused on leaf development in L. chinense. To date, some progress has been made in this field. For instance, a comparison of L. chinense leaf transcripts among various developmental stages has been performed (Ma et al., 2018). Ectopic expression of KNOX6 from L. chinense in A. thaliana was found to result in numerous lobed leaves. It has been proposed that LcKNOX6 might participate in leaf development in L. chinense (Ma et al., 2020). Also, some LcAP2/ERFs also seems involved in leaf morphogenesis through STC analysis in various tissues and the further anatomical assay of the leaf bud (Zong et al., 2021).

CUC, a member of the NAC transcription factor family with a highly conserved N-terminal domain and a variable C-terminal domain (Taoka et al., 2004), was first isolated from A. thaliana (Aida et al., 1997). Leaves usually originate from protuberant clusters of cells around the shoot apical meristem (SAM) (Kessler & Sinha, 2004). Experimental evidence suggests important roles of CUC2 in SAM formation and organ boundary formation, with cuc1 cuc2 double mutants exhibiting a single cup-shaped cotyledon and a deficiency in embryonic SAM (Aida et al., 1997). Moreover, CUC transcription factors are commonly called plant dissectors partially due to their functions in leaf margin development (Bilsborough et al., 2011). Early reports showed that the expression of the CUC2 gene is modulated by auxin (Vernoux et al., 2000; Aida et al., 2002; Furutani et al., 2004). PINFORMED 1 (PIN1), as the main auxin transporter (Galweiler et al., 1998), is regulated by CUC2 during serration formation (Kawamura, Horiguchi & Tsukaya, 2010). Auxin, PIN1, and CUC2 have been demonstrated to compose a regulatory network involved in leaf serration formation (Bilsborough et al., 2011). Belonging to a class of plant-specific microRNAs, miR164 is a significant regulatory factor for normal plant development (Mallory et al., 2004). It has been confirmed that CUC2 is the target gene of miR164 (Rhoades et al., 2002). miR164 negatively regulates CUC2, and their balance influences the depth of leaf lobation (Nikovics et al., 2006). In addition, DEVELOPMENT-RELATED PcG TARGET IN THE APEX4 (DPA4) inhibits leaf serration formation by negatively regulating CUC2 expression independent of miR164 (Engelhorn et al., 2012). Investigators have proven that NGATHA-LIKE 1 (NGAL1) directly inhibits CUC2 expression by binding to the CUC2 promoter and thereby negatively regulates the formation of leaf margin serration (Shao et al., 2020). In addition, cuc2 cuc3 double mutants exhibit a lack of an axillary meristem (Hibara et al., 2006), indicating that CUC2 might be involved in axillary meristem initiation. Previous studies have suggested that CUC2 plays important roles in adventitious shoot formation (Daimon, Takabe & Tasaka, 2003) and floral organ development (Aida et al., 1997; Baker et al., 2005; Kamiuchi et al., 2014; Gonzalez-Carranza et al., 2017). Moreover, CUC2 plays a role in internode development, with BpCUC2 overexpression leading to internode shortening in Betula pendula (Liu et al., 2019). Taken together, these observations suggest that CUC2 participates in both growth and development in plants.

In this study, LcCUC2L and its promoter were isolated from L. chinense and characterized by conducting bioinformatics analysis. Assessments of the functions of LcCUC2L in transgenic A. thaliana plants were performed using overexpression assays. As a result, GUS staining revealed that LcCUC2L was expressed in the cotyledons of A. thaliana seedlings, which indicated that LcCUC2L may play a role in cotyledon development. Intriguingly, almost all the transgenic A. thaliana lines displaying abnormal cotyledons and an increased rosette leaf number. Further SEM observation and venation pattern analysis showed that overexpression of LcCUC2L induced abnormal lamina epidermis cells and defective vascular tissue of cotyledons. Hormone determination and RT-qPCR results indicated that LcCUC2L affects leaf development by regulating the auxin content and the expression of genes involved in auxin synthesis, transport, and leaf shape development in A. thaliana. Our results will help to reveal the molecular mechanisms of leaf development in L. chinense.

Materials & Methods

Plant materials, growth conditions, and treatments

Samples of leaves at different developmental stages (Tu et al., 2019) were harvested from an adult L. chinense tree in Xiashu, Jurong County, Jiangsu Province, China (119°13′20′E, 32°7′8′N) (Yang et al., 2014). Plant materials were maintained at −80 °C prior to analysis.

Wild-type (WT) A. thaliana (Columbia-0), transgenic A. thaliana, Nicotiana benthamiana (Ben) plants, and L. chinense seedlings were cultivated at 23 °C under a long-day photoperiod (16 h light, 8 h dark) in a growth chamber (70% relative humidity).

For the IAA treatment, L. chinense seedlings grown in the incubator for 5 months were sprayed with 200 µM IAA on leaves and then sampled at 48 h. Water was used as the control. The samples were placed in an ultra-low-temperature freezer for further analysis.

DNA, RNA extraction, and RT-qPCR

A DNAsecure Plant Kit (Tiangen) was used for DNA isolation from L. chinense. RNA was extracted from L. chinense and A. thaliana leaves using the RNAprep Pure Plant Kit (Tiangen). Subsequently, first-strand cDNA was synthesized using PrimeScript™ RT Master Mix (TaKaRa) in accordance with the manufacturer’s protocol. The LcCUC2L transcript levels in leaves of different developmental stages (Tu et al., 2019) were investigated by RT-qPCR. Moreover, we also detected LcCUC2L expression levels under IAA treatment. L. chinense Actin 97 was used as the internal control. In addition, the transcription levels of LcCUC2L and some other genes in WT and transgenic lines were examined using specific primers (Table S1). A. thaliana Actin 2 was used as the internal quantitative control. RT-qPCR was carried out using the SYBR Premix EX Taq kit (TaKaRa). We applied the 2−ΔΔCT method to analyze the data from relative quantification (Livak & Schmittgen, 2001).

Cloning and bioinformatics analysis

Based on the known transcriptome database (Ma et al., 2018), the CUC2 gene was identified in L. chinense. The full-length LcCUC2L gene was obtained using RT-PCR and RACE. The LcCUC2L gene was assembled with the middle region, 5′ sequence and 3′ sequence. ORF Finder was used for prediction of the open reading frame (ORF) and protein sequence. The structure of the LcCUC2L gene was analyzed by the online software GSDS (http://gsds.gao-lab.org/). Analysis of the physicochemical characteristics of the proteins was conducted online at the following website: (https://web.expasy.org/protparam/). PredictProtein was used to forecast protein secondary structure. Subcellular localization was predicted by applying WoLF PSORT. Protein conserved domain analysis of LcCUC2L was performed with NCBI (https://www.ncbi.nlm.nih.gov/cdd/?term). The amino acid sequences of the CUC2 gene in other plant species were acquired from GenBank (http://www.ncbi.nlm.nih.gov/genbank). Multiple sequence alignment of LcCUC2L and its homologous genes was performed with ClustalW (https://www.genome.jp/tools-bin/clustalw). The results were uploaded to ESPript 3.0 (http://espript.ibcp.fr/ESPript/cgi-bin/ESPript.cgi). A phylogenetic tree was constructed using MEGA 7.

After consulting the published L. chinense genome data (Chen et al., 2019), approximately 2 kb of genomic DNA upstream of the start codon of LcCUC2L was cloned. The PlantCARE (http://bioinformatics.psb.ugent.be/webtools/plantcare/html) was used to predict the cis-acting elements of ProLcCUC2L.

Subcellular localization of LcCUC2L

The whole coding sequence (CDS) of LcCUC2L without a stop codon was ligated to a linearized pBI121-eGFP vector. The recombinant vector was transformed into Agrobacterium tumefaciens GV3101 (ShiFeng). And the 35S::eGFP was used as the control. Positive clones and the P19 vector were cultured in Luria-Bertani (LB) medium at 28 °C. Next, the cells were collected and suspended in buffer containing 10 mM MgCl2 and 150 μM acetosyringone (AS). Two bacterial suspensions were mixed and injected into tobacco leaves (Sparkes et al., 2006). After 48 h of injection, the tobacco leaves were stained with the nuclear dye 4′, 6-diamidino-2-phenylindole (DAPI) to visualize the nucleus. Subsequently, the fluorescent signals in the epidermis cells of tobacco leaves were examined with a confocal laser scanning microscope (LSM 710, Zeiss, Germany).

Plasmid construction and genetic transformation

The full-length ORF of LcCUC2L was cloned into the linearized overexpression vector pBI121. Similarly, the sequence of ProLcCUC2L was inserted at the HindIII/NcoI site of the pCAMBIA1301 vector by substituting the CaMV35S promoter, yielding the construct ProLcCUC2L::GUS. The Agrobacterium-mediated floral dip method was used for the genetic transformation of A. thaliana (Clough & Bent, 1998). Transgenic A. thaliana seeds were screened on 12 MS medium containing antibiotics. Using leaf genomic DNA as templates, T1-positive transgenic plants were identified by PCR using specific primers. We acquired T3 transgenic plants through continuous screening for three generations.

Venation pattern analysis and scanning election microscopy

To compare cotyledon vein patterns of transgenic and WT A. thaliana plants, the cotyledons were bleached with 75% (v/v) ethanol and observed using stereomicroscopy. For cotyledon epidermal cell analysis, the cotyledons of transgenic and WT A. thaliana plants were fixed in FAA fixative solution (38% formalin, 50% ethanol, and glacial acetic acid). After critical point drying (CPD), the samples were coated with an Edwards E-1010 ion sputter golden coater and examined with a scanning electron microscope (FEI Quanta 200 FEG MKII).

Auxin content measurement

Basal rosette leaves samples of 0.6 g were harvested from 3-week-old WT and LcCUC2L transgenic A. thaliana plants and immediately frozen in liquid nitrogen. Determination of indole-3-acetic acid (IAA) in A. thaliana leaves was performed by ELISA (Huding, Shanghai, China). Three replicates were used for each sample.

Histochemical staining to detect GUS activity

The β-Galactosidase Reporter Gene Staining Kit (Leagene) was used for GUS histochemical staining in ProLcCUC2L transgenic and WT A. thaliana plants. In this study, seven-day-old seedlings and fourteen-day-old seedlings were processed. Moreover, leaves at different stages of development (10, 20 and 30 days of age) from plants planted in soil were subjected to GUS staining. All materials were treated with GUS staining solution and incubated overnight at 37 °C, followed by bleaching with 75% (v/v) ethanol.

Statistical analysis

Both the control and the experimental lines had three repetitions. The data are expressed as the mean ± standard deviation (SD). Statistical significance was determined with Student’s t-test. P < 0.05 was regarded as significant (*), and P ≤ 0.01 was considered highly significant (**).

Results

Cloning and sequence analysis of LcCUC2L and ProLcCUC2L

The full length of the LcCUC2L gDNA sequence is 4,374 bp and consists of three exons and two introns (Fig. 1A). LcCUC2L (accession number: lcCUC2MW629054) contains a 1,017 bp ORF (Data S1), which encodes 338 amino acids and has a predicted molecular weight of 37.32 kDa and a theoretical PI of 8.45. The instability index was 44.46, indicating that the protein is unstable. The secondary structure of LcCUC2L was predicted to consist of helix (3.25%), loop (84.02%) and strand (12.72%) structures. Subcellular localization prediction showed that the LcCUC2L protein was mainly distributed in the cell nucleus. The amino acid sequences of the LcCUC2L protein in L. chinense and other plants were analyzed via multiple sequence alignment. Furthermore, the amino acid sequence of LcCUC2L was compared with the sequences of other CUC2 proteins in Arabidopsis lyrata subsp. lyrata, A. thaliana, Ananas comosus, Cardamine hirsute, Glycine max, and Zea mays. Similar to the N-terminus of other CUC2 proteins, the N-terminus of the LcCUC2L protein contains a conserved NAC domain, with five conserved subdomains (A, B, C, D, E), indicating that these proteins belong to the CUC subfamily of the NAC family. In addition, CUC genes harbor a variable CTD domain. The K motif (PSKKTKVPSTIS), V motif (EHVSCFS), L motif (SLPPL) and W-motif (WNY) can be observed in CUC2. In contrast, CUC1 does not have a K motif and there is only an L motif in CUC3 (Larsson et al., 2012). LcCUC2L harbors a variable CTD domain, and the K motif, L motif, V motif and W motif were observed in the LcCUC2L protein (Fig. 1B). The V motif of CUC1 and CUC2 has been identified as a miR164 recognition site (Rhoades et al., 2002). As shown in Fig. 1C, which indicated the LcCUC2L might be recognized by miR164. Moreover, phylogenetic tree analysis confirmed that CUC proteins could be classified into three clads: CUC1, CUC2, and CUC3. The LcCUC2L protein is most closely related to the Ananas comosus CUC2 protein (Fig. 1D). In conclusion, these results show that the sequence is highly homologous to CUC2, so we named it LcCUC2L.

Figure 1 Sequence analysis of LcCUC2L.

(A) LcCUC2L gene structure diagram. (B) Homology analysis of the LcCUC2L protein. Amino acid sequence alignment of LcCUC2L and CUC2 proteins in other plants. AlCUC2: Arabidopsis lyrata subsp. lyrata (XP_002866009.1), AtCUC2: Arabidopsis thaliana (OAO90779.1), ChCUC2: Cardamine hirsuta (ACL14370.1), GmCUC2: Glycine max (XP_003541838.1), AcCUC2: Ananas comosus (OAY67566.1), and ZmCUC2: Zea mays (PWZ16696.1). (C) Nucleotide sequence alignment of the LcCUC2L and AtCUC2 that targeted by miR164. (D) Phylogenetic analysis of LcCUC2L and CUC proteins from other plants. EsCUC1: Eutrema salsugineum (XP_006407005.1), ChCUC1: Cardamine hirsute (ACL14369.1), CrCUC1: Capsella rubella (XP_006296616.1), AlCUC1: Arabidopsis lyrata subsp. lyrata (XP_002882916.1), AtCUC1: Arabidopsis thaliana (BAB20598.1), AlCUC2: Arabidopsis lyrata subsp. lyrata (XP_002866009.1), AtCUC2: Arabidopsis thaliana (OAO90779.1), ChCUC2: Cardamine hirsuta (ACL14370.1), GmCUC2: Glycine max (XP_003541838.1), AcCUC2: Ananas comosus (OAY67566.1), ZmCUC2: Zea mays (PWZ16696.1), AtCUC3: Arabidopsis thaliana (AAP82630.1), GmCUC3: Glycine max (XP_003523523.1), StCUC3: Solanum tuberosum (NP_001275002.1), ChCUC3: Cardamine hirsuta (ACL14365.1), and AcCUC3: Aquilegia coerulea (ACL14364.1).

To explore the function of LcCUC2L, a 2,028 bp (Data S2) sequence upstream of the translational initiation site (ATG) of the LcCUC2L gene, which contains the promoter sequence of LcCUC2L, was amplified using the DNA template of L. chinense. PlantCARE software was used to analyze the cis-acting elements of ProLcCUC2L. As shown in Table 1, some cis-acting elements are involved in light regulation, such as AE-box, ATCT-motif, Box4, G-Box, G-box, I-box, TCCC-motif, and chs-CMA1a. LTR and WUN motifs associated with low-temperature responsiveness and wound responsiveness, respectively, exist in the sequence. In addition, some hormone-related elements were identified, including the abscisic acid responsive regulatory motif (ABRE), auxin-responsive element (TGA-element) and MeJA-responsiveness elements (CGTCA-motif and TGACG-motif). The predicted results revealed that the core promoter elements of ProLcCUC2L contained TATA-box and CAAT-box.

Table 1 Details of cis-acting elements of the ProLcCUC2L in L. chinense.

Name	Number	Sequence	Function	
ABRE	3	CACGTG/ ACGTG	cis-acting element involved in the abscisic acid responsiveness	
AE-box	1	AGAAACAA	part of a module for light response	
ARE	2	AAACCA	cis-acting regulatory element essential for the anaerobic induction	
ATCT-motif	1	AATCTAATCC	part of a conserved DNA module involved in light responsiveness	
Box 4	1	ATTAAT	part of a conserved DNA module involved in light responsiveness	
CAAT-box	27	CAAT/CAAAT	common cis-acting element in promoter and enhancer regions	
CGTCA-motif	3	CGTCA	cis-acting regulatory element involved in the MeJA-responsiveness	
G-Box	2	CACGTGAAA/ CACGTG	cis-acting regulatory element involved in light responsiveness	
G-box	2	CACGTG/CACGTC	cis-acting regulatory element involved in light responsiveness	
I-box	1	atGATAAGGTC	part of a light responsive element	
LTR	1	CCGAAA	cis-acting element involved in low- temperature responsiveness	
TATA-box	8	TATAA/TATA/ ccTATAAAaa/TATACA	core promoter element around -30 of transcription start	
TCCC-motif	1	TCTCCCT	part of a light responsive element	
TGA-element	1	AACGAC	auxin-responsive element	
TGACG-motif	3	TGACG	cis-acting regulatory element involved in the MeJA-responsiveness	
chs-CMA1a	1	TTACTTAA	part of a light responsive element	
WUN-motif	1	AAATTTCCT	wound-responsive element	

Expression pattern analysis of LcCUC2L

The transcript levels of LcCUC2L were determined by RT-qPCR in leaves of different developmental stages (Fig. 2A). The highest expression levels of LcCUC2L were observed in the leaf bud, and the lowest abundance of LcCUC2L transcripts were in mature leaves (Fig. 2B, Table S2). The results indicated that LcCUC2L may play an important role in leaf bud development.

Figure 2 Expression pattern analysis of LcCUC2L.

(A) Leaves of four different developmental stages in L. chinense. S1 leaves: leaf bud; S2 leaves: newly expanded leaves; S3 leaves: larger leaves; S4 leaves: mature leaves, bars = two cm. (B) Transcript levels of LcCUC2L in leaves of different developmental stages in L. chinense. (C) Expression level of LcCUC2L under 200 µM IAA. (D) Histochemical analysis of GUS in transformed A. thaliana leaves, bars = 0.1 cm. (E) Subcellular localization of LcCUC2L protein, bars = 2,000 nm. Note: In B and C data are shown as the mean ± SD, B and C data are based on three repetitive experiments. **P < 0.01 and *P < 0.05, Student’s t-test.

Based on cis-element sequence analysis of the LcCUC2L promoter, we predicted that LcCUC2L might be associated with auxin. To confirm the effect of auxin on LcCUC2L expression, we sprayed 200μM IAA on the leaves of L. chinense. The application of IAA significantly inhibited the expression of LcCUC2L (Fig. 2C, Table S3), which suggested that IAA represses LcCUC2L expression.

To further assess the expression pattern of LcCUC2L, ProLcCUC2L was fused to the GUS reporter and subsequently transferred into A. thaliana (hereafter ProLcCUC2L). After screening in media containing hygromycin and DNA detection, we obtained 8 transgenic lines (Fig. S1). Histochemical GUS staining was applied to detect the expression levels of the GUS gene in transgenic A. thaliana plants, which displayed inducible activity of ProLcCUC2L. In the 7-day-old and 14-day-old transgenic seedlings, GUS staining was mainly detected in the cotyledon lamina and roots (Fig. 2D). In contrast, low GUS activity was observed in the cotyledon petiole and hypocotyl of the transgenic seedlings. GUS activity was not detected in the true leaves of 14-day-old transgenic seedlings. These results indicated that LcCUC2L may have participated in cotyledon development in the transgenic A. thaliana seedlings.

Confocal images with DAPI nuclear staining (blue) were taken 48 h after transfection, showing GFP (green) expression that indicates the subcellular localization of LcCUC2L. Figure 2E shows that the green fluorescence of the control 35S::GFP was distributed throughout the cells. In contrast, the green fluorescence of the 35S::LcCUC2L-eGFP fusion protein was observed in the nucleus, suggesting that LcCUC2L localized to the nucleus. This result was fully consistent with the subcellular localization result of the AtCUC2 protein in A. thaliana (Taoka et al., 2004).

Overexpression of LcCUC2L regulates leaf development in A. thaliana

To investigate the functions of LcCUC2L, we obtained transgenic A. thaliana plants overexpressing LcCUC2L under the control of the CaMV 35S promoter. After screening on selective medium, seven positive transgenic plants were acquired. Subsequently, three high expression lines with the same phenotypes (OE 1, OE 3, and OE 5) were selected for phenotype analysis (hereafter 35S::LcCUC2L-OE1, 35S::LcCUC2L-OE3, and 35S::LcCUC2L-OE5) (Fig. 3F, Table S4). We found that true leaves began to emerge on or near the 7th day, as shown in Fig. 3A. The results revealed significant variation in the cotyledon phenotypes between WT and 35S::LcCUC2L plants. In contrast to WT cotyledons, the cotyledons of the 35S::LcCUC2L plants appeared as long strips and without petioles, and they had narrower blades compared to the WT cotyledons. Moreover, we compared vascular development between the WT and 35S::LcCUC2L cotyledons. Two kinds of veins exist in WT cotyledons: primary veins and subprime veins. Several subprime veins branch from the midvein and then unite to form areoles separated by veins (Figs. 3B, 3C) (Sieburth, 1999). In contrast, 35S::LcCUC2L plant cotyledons displayed an incomplete vascular pattern and had only a single central strand (Figs. 3B, 3C). In conclusion, the 35S::LcCUC2L cotyledons exhibited significant deficiencies in vascular development. To examine the cotyledon cell changes in the transgenic plants, we imaged the epidermal cells of 35S::LcCUC2L and WT with scanning election microscopy (SEM). Scanning election microscopy observations revealed that in WT cotyledons, the epidermal cells were organized in a specific pavement-like pattern (Fig. 3D) (Koyama et al., 2007). Remarkably, the shape and arrangement of the epidermal cells in the 35S::LcCUC2L cotyledons were similar to those of the WT petiole, with both being arranged regularly and exhibiting a rectangular shape. Taken together, the findings revealed large variations in cotyledon morphology between the 35S::LcCUC2L and WT A. thaliana plants, demonstrating that LcCUC2L affects cotyledon development in A. thaliana.

Figure 3 The phenotypes of the LcCUC2L overexpression transgenic lines and WT A. thaliana plants.

(A) Seven-day-old cotyledons of WT and 35S::LcCUC2L plants, bars = 0. 1 cm. (B) Venation patterns of WT and 35S::LcCUC2L plants, bars = 0.1 cm. (C) Illustrations of the vascular patterns of 7-day-old cotyledons of WT and 35S::LcCUC2L plants. (D) SEM images of cotyledons of WT and 35S::LcCUC2L plants. (E) 30-day-old WT and 35S::LcCUC2L A. thaliana, bars = 2 cm. (F) qRT-PCR analysis of LcCUC2L expression in WT and transgenic plants. Values are means ± SD (n = 3). Data are based on three repetitive experiments. **P < 0.01 and *P < 0.05, Student’s t-test. (G) The rosette leaves of WT and 35S::LcCUC2L A. thaliana, bars = 2 cm. (H) The number of rosette leaves in WT and 35S::LcCUC2L A. thaliana. Values are means ± SD (n = 18). **P < 0.01 and *P < 0.05, Student’s t-test.

Ten days later, we transplanted seedlings of the homozygous transgenic lines and WT A. thaliana plants into nutrient soil. We observed that the rosette leaves of the 30-day-old 35S::LcCUC2L plants grew in clusters and were small and numerous (Figs. 3E, 3G). The 35S::LcCUC2L-OE1 and 35S::LcCUC2L-OE3 had 20 rosette leaves on average and the 35S::LcCUC2L-OE5 had 18 rosette leaves on average (Table S5), which was considerably more than the number in WT (Fig. 3H). These results indicated that LcCUC2L influences leaf development in A. thaliana.

LcCUC2L regulates leaf development by upregulating the expression of some genes related to auxin and leaf shape development

Many recent investigations have indicated that auxin plays a vital role in leaf development (Xiong & Jiao, 2019). The 35S::LcCUC2L plants amassed more IAA than the WT A. thaliana plants (Fig. 4A, Table S6). To explore whether LcCUC2L regulates leaf development by affecting auxin signaling, the expression levels of auxin biosynthesis genes and auxin transport genes were examined (Table S7). The transcript levels of the auxin biosynthetic genes YUCCA (AtYUC2 and AtYUC4) (Fig. 4B), an auxin influx carrier (AtAUX1), and efflux carriers (AtPIN1, AtPIN3, and AtPIN4) (Fig. 4C) were increased in LcCUC2L-expressing plants compared to WT A. thaliana plants. However, LcCUC2L overexpression had little effect on the expression level of AtYUC6. The results imply that the effects of LcCUC2L in leaf development might be enhanced by changes in auxin biosynthesis and polar transport. Additionally, we analyzed some genes related to leaf shape development, i.e., KNAT1, KNAT2, KNAT6 (for KNOTTED-like from A. thaliana), SHOOOTMERISTEMLESS (STM), and DPA4. We found that 35S::LcCUC2L plants presented much higher transcript levels of KNAT6, KNAT2, DPA4, and AtCUC2 than WT A. thaliana plants (Fig. 4D). The results indicate that the overexpression of LcCUC2L led to the upregulation of some genes related to leaf shape development and thus affected leaf development in A. thaliana.

Figure 4 LcCUC2L promoted auxin accumulation in leaves and upregulated the expression of some genes in transgenic A. thaliana plants (35S::LcCUC2L).

(A) Total IAA content in the rosette leaves of three-week-old seedlings of WT and 35S::LcCUC2L A. thaliana plants. (B) Transcript levels of auxin biosynthetic genes (AtYUC2, AtYUC4, and AtYUC6) in WT and 35S::LcCUC2L A. thaliana plants. (C) Transcript levels of auxin influx carriers (AtAUX1) and efflux carriers (AtPIN1, AtPIN3, and AtPIN4) in WT and 35S::LcCUC2L A. thaliana plants. (D) Transcript levels of some genes involved in the leaf shape development (AtKNAT2, AtKNAT6, AtDPA4, and AtCUC2) in WT and 35S::LcCUC2L A. thaliana plants. Note: In A, B, C, and D data are shown as the mean ± SD, based on three repetitive experiments. **P < 0.01 and *P < 0.05, Student’s t-test.

Discussion

Leaf shape is an important plants traits. Early studies of leaf shape development have focused on model plants, but there is still limited in woody plants. L. chinense has a distinctive leaf shape (Fig. 2A), which provides a wealth of reference information to uncover the molecular mechanisms of leaf shape development. Based on a comparison of L. chinense leaf transcripts among different developmental stages, the CUC2 gene was identified as a candidate gene related to leaf development (Ma et al., 2018). Subsequently, we identified the full-length sequences of LcCUC2L and its promoter isolated from L. chinense and analyzed their functions via overexpression analysis.

In this study, we discovered that overexpression of LcCUC2L strongly affected cotyledon development in transgenic A. thaliana, resulting in long, narrow cotyledons without petioles (Fig. 3A) but with abnormal epidermal cells (Fig. 3D). In the early stages of leaf organogenesis, the proximal–distal, medial–lateral and adaxial–abaxial axes are established (Tsukaya, 2006; Kidner & Timmermans, 2007). The proximal–distal axis domains of WT A. thaliana leaves are composed of the lamina and the petiole (Jover-Gil et al., 2012). In the present study, abnormal epidermal cells were present in the cotyledons of the 35S::LcCUC2L plants, with those in the lamina resembling the epidermal cells of the wild-type petiole (Fig. 3D). Previous studies have reported blade growth on the petiole and a diminished proximal domain in the bop1 bop2 double mutant (Ha et al., 2007). In addition, the as2 mutation causes the petiole to curl upward and to sometimes produce leaflets (Iwakawa et al., 2007). These phenotypes are associated with the ectopic expression of meristem class I homeobox KNOX genes, i.e., KNAT2, KNAT6, STM and BP (Ha et al., 2003; Iwakawa et al., 2007; Ikezaki et al., 2010; Jun, Ha & Fletcher, 2010). To our surprise, overexpression of LcCUC2L in A. thaliana promoted the expression of KNAT2 and KNAT6 (Fig. 4D). Further analyses suggested that the ectopic expression of LcCUC2L regulated the proximal–distal axis by enhancing the expression of KNAT2 and KNAT6; thus, the cotyledons of 35S::LcCUC2L A. thaliana tended to develop as long, narrow cotyledons without petioles and exhibited abnormal epidermal cells.

Furthermore, cotyledon vein development in 35S::LcCUC2L plants was defective (Figs. 3B, 3C). It is widely known that auxin plays an indispensable role in vascular development (Aloni, 1995). Mutations in some genes related to the auxin signaling pathway (AUXIN RESPONSE FACTOR 5 (ARF5), AUXIN RESISTANT 6 (AXR6) and IAA12) contribute to incomplete vascular development (Przemeck et al., 1996; Hobbie et al., 2000; Hamann et al., 2002). Moreover, evidence suggests that polar auxin transport plays a significant role in determining leaf vascular patterns. Studies have shown that the expression of CUC genes is regulated by PIN1 (Aida et al., 2002) and that blocking PIN-mediated polar auxin transport (PAT) results in cotyledon fusion (Liu, Xu & Chua, 1993). It has been proposed a feedback regulation network of leaf margin development among auxin, PIN1, and CUC2 (Maugarny et al., 2016). The auxin content and PIN1 transcript level of 35S::LcCUC2L plants were higher than those of WT A. thaliana plants in this study (Figs. 4A, 4C). Bioinformatic analysis revealed that the auxin-responsive element is present in ProLcCUC2L (Table 1). The authors identified three upstream transcription factors that bind to the auxin response element (CACATG) of the BpCUC2 promoter (Liu et al., 2018). Our results revealed that the expression of LcCUC2L markedly decreased under IAA treatment (Fig. 2C). These findings imply that LcCUC2L may be related to the metabolism of auxin. Based on the above results, we speculate that LcCUC2L elevated auxin content by upregulating local auxin biosynthesis and polar transport, which affected cotyledon development. Moreover, GUS activity was detected in the cotyledons of transgenic A. thaliana seedlings (Fig. 2D). This finding is in line with the fact that LcCUC2L regulates cotyledon development in transgenic A. thaliana plants.

Ectopic expression of LcCUC2L produced numerous rosette leaves (Figs. 3G, 3H). Leaves are lateral organs that develop continuously at the flanks of the SAM in flowering plants (Kessler & Sinha, 2004). The cotyledons of cuc1 cuc2 double mutant seedlings are fused into a single cup-shaped cotyledon without a SAM (Aida et al., 1997). In addition, the KNOX gene is required to maintain the development and function of the SAM (Scofield, Dewitte & Murray, 2014). Overexpression of LcCUC2L led to upregulated expression of KNAT2 and KNAT6 (Fig. 4D), indicating that LcCUC2L may regulate SAM development by affecting the expression of KNAT2 and KNAT6. Moreover, LcCUC2L was prominently expressed in the leaf buds (Fig. 2A), which is in accordance with BpCUC2 having the highest expression level in buds (Liu et al., 2018), suggesting that LcCUC2L may function in the early stage of leaf development. SAM and leaf tissue can be observed in the leaf buds of L. chinense (Ma et al., 2018). We assumed that the development of the SAM was influenced by the overexpression of LcCUC2L, which increased the number of rosette leaf. These findings call for further research to verify this possibility.

Research has demonstrated that the CUC1 and CUC2 genes descended from a common ancestor that has undergone two duplications and sequence loss events in the evolutionary process (Bowers et al., 2003). Previous data have shown no obvious divergence between CUC2 and the ancestral gene, while CUC1 has diverged from the ancestral gene, resulting in a different function (Hasson et al., 2011). For example, both CUC1 and CUC2 participate in primordium development, while CUC2 and CUC3 were proven to be involved in leaf margin development (Nikovics et al., 2006; Hasson et al., 2011). Moreover, the phylogenetic tree showed that LcCUC2L and AtCUC1 were not clustered in one branch (Fig. 1D). We found that LcCUC2L had the nearest evolutionary distance to AcCUC2 (Fig. 1D) and the amino acid sequence similarity of LcCUC2L and AtCUC2 was 52.36%. These results indicated that there may be functional differentiation between LcCUC2L, AtCUC2 and AtCUC1.

Studies in A. thaliana have suggested that cuc1 cuc2 double mutants showed cotyledon fusion and defective SAMs, revealing a key role for CUC genes in cotyledon development and SAM formation (Aida et al., 1997). However, our study showed that overexpression of LcCUC2L strongly conferred long, narrow cotyledons without petioles (Fig. 3A) and abnormal epidermal cell (Fig. 3D) phenotypes. Moreover, LcCUC2L showed strong expression in cotyledons (Fig. 2D), which further confirmed that LcCUC2L might play a role in regulating cotyledon development. In A. thaliana seedlings, AtCUC2 was expressed in the boundaries between the cotyledons, the first true leaves, and the SAM. This finding is in line with CUC2 being a boundary-specific gene (Larsson et al., 2012; Takada et al., 2001). During leaf development, GUS activity tended to become weaker in the sinus of the leaves, which indicated that local inhibition of the CUC2 gene may cause serration (Nikovics et al., 2006). A. thaliana has serrated leaves, and a smooth leaf margin phenotype was observed in the cuc2 mutant, suggesting that CUC2 participates in leaf serration in A. thaliana (Hasson et al., 2011). Accordingly, it can be inferred that the expression pattern of the AtCUC2 gene is closely related to its function. However, the LcCUC2L gene was highly expressed in the cotyledons and affected the cotyledon development. Moreover, the ProLcCUC2-GUS line showed strong expression in the roots as well as cotyledons (Fig. 2D). This phenomenon interested us, and the role of LcCUC2L in root development might our next research direction.

CUC1 was detected in the whole area of the cotyledons. 35S::AtCUC1 transgenic plants showed small lobed cotyledons with defective vasculature (Hibara, Takada & Tasaka, 2003). The phenotype was similar to that of the LcCUC2L overexpressing plants. Both play role in cotyledon development. In the cotyledons of 35S::AtCUC1, adventitious SAMs were observed on the adaxial surface of this region (Hibara, Takada & Tasaka, 2003). To date, we have not found that LcCUC2L could induce adventitious SAMs in transgenic A. thaliana plants. The above results show that the LcCUC2L and CUC genes seem to be homologous in sequence, but non-homologous in function. Moreover, differences in regulatory mechanisms exist between herbaceous and woody plants, and the expression of LcCUC2L in L. chinense may yield different results than that in A. thaliana. Therefore, to test the above hypothesis, it is necessary to transfer the LcCUC2L gene into L. chinense.

In summary, these observations suggest that LcCUC2L may play a critical role in leaf shape development in L. chinense. Our results lay a foundation for future studies of the mechanisms of leaf shape development and provide initial insight into the functions of the CUC2 gene in L. chinense.

Conclusions

In this study, histochemical GUS staining revealed that LcCUC2L was expressed in the cotyledons of A. thaliana seedlings, which indicated that LcCUC2L may play a role in cotyledon development. Ectopic expression of LcCUC2L resulted in long, narrow cotyledons without petioles and increased rosette leaf number. Further analysis showed that overexpression of LcCUC2L induced abnormal lamina epidermis cells and defective vascular tissue in cotyledons. Hormone determination and RT-qPCR results indicated that LcCUC2L affects leaf development by regulating the auxin content and the expression of genes related to auxin and leaf shape development. These findings indicate that LcCUC2L may influence leaf development in L. chinense and provide insights into the regulatory mechanisms of leaf development in L. chinense.

Supplemental Information

Supplemental Information 1 The ORF sequences of LcCUC2L

Click here for additional data file.

Supplemental Information 2 The promoter sequence of LcCUC2L

Click here for additional data file.

Supplemental Information 3 The DNA detection of T1 transgenic lines and WT A. thaliana.

Click here for additional data file.

Supplemental Information 4 All primers in this study

Click here for additional data file.

Supplemental Information 5 Relative expression levels of LcCUC2L in leaves of different developmental stages in L. chinense.

We applied the 2T−ΔΔC method to analyze the data from relative quantification.

Click here for additional data file.

Supplemental Information 6 The expression levels of LcCUC2-like under 200 µM IAA treatment in L. chinense.

The expression levels of LcCUC2-like under 200 µM IAA treatment in L. chinense. We applied the 2T−ΔΔC method to analyze the data from relative quantification

Click here for additional data file.

Supplemental Information 7 The expression levels of LcCUC2-like in WT and in dependent transgenic lines

We applied the 2T−ΔΔC method to analyze the data from relative quantification.

Click here for additional data file.

Supplemental Information 8 The number of rosette leaves in WT and transgenic A. thaliana plants

Click here for additional data file.

Supplemental Information 9 The IAA content of WT and transgenic A. thaliana plants

Click here for additional data file.

Supplemental Information 10 The expression level of some genes in WT and transgenic A. thaliana plants

We applied the 2T−ΔΔC method to analyze the data from relative quantification.

Click here for additional data file.

The authors thank their laboratory colleagues for their help in lab work, including Huanhuan Liu, Yufang Shen, Zhonghua Tu, Hui Xia, Xi Wang, Mujun Liu, Shenhua Zhu, Chengge Zhang, Lichun Yang, Xinyu Zhai, Shan Hu, and Xujia Wu.

Additional Information and Declarations

Competing Interests

Author Contributions

DNA Deposition

Data Availability

The authors declare there are no competing interests.

Shaoying Wen conceived and designed the experiments, performed the experiments, analyzed the data, prepared figures and/or tables, authored or reviewed drafts of the paper, and approved the final draft.

Jiayu Li performed the experiments, analyzed the data, prepared figures and/or tables, authored or reviewed drafts of the paper, and approved the final draft.

Ziyuan Hao performed the experiments, prepared figures and/or tables, and approved the final draft.

Lingmin Wei performed the experiments, prepared figures and/or tables, took part in the seeds of transgenic A. thaliana collection, and approved the final draft.

Jikai Ma and Yaxian Zong performed the experiments, prepared figures and/or tables, was involved in the plant materials collection, and approved the final draft.

Huogen Li conceived and designed the experiments, prepared figures and/or tables, authored or reviewed drafts of the paper, and approved the final draft.

The following information was supplied regarding the deposition of DNA sequences:

The sequences described here are available in the Supplemental Files and at GenBank: LcCUC2 MW629054.

The following information was supplied regarding data availability:

The raw data are available in the Supplementary Files.

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
