# Peer review of "Overexpression of the LcCUC2-like gene in Arabidopsis thaliana alters the cotyledon morphology and increases rosette leaf number"

_PeerJ, doi:10.7717/peerj.12615_

## Round 0.1 · original submission · Major Revisions

In addition to the questions/suggestions raised by the two reviewers, I am concerned that because of the experimental approach used here it is difficult to make any convincing conclusions. It is difficult to determine if LcCUC2 is functionally homologous to the Arabidopsis counterparts because a complementation of the At cuc deficient mutant hasn't been done. Should it not be possible to complement a cuc1/cuc1;cuc2/+ line? This would at least suggest some functional analogy. In addition, AtCUC2 overexpressors should be compared with the LcCUC2 at the same developmental stages. The expression pattern of the LcCUC2 should also be compared to the expression pattern of the endogenous Arabidopsis gene. Does the overexpression of LcCUC2 affect the expression of AtCUC2 gene? The ProLcCUC2-GUS line shows strong expression in the roots as well as leaves. Is this pattern similar with the AtCUC2 gene? There is a list of predicted cis-acting elements in Table 1, what the functional significance for any of these? The response to different treatments should be tested with the GUS line.

ddreb

Reviewer 1 ·

Basic reporting

For Table_S3 (Number of rosette leaves), only the average and the standard deviation is provided. Please include all data.

Experimental design

no comment

Validity of the findings

no comment

Additional comments

Functional identification of the LcCUC2 gene from Liriodendron chinense in leaf development

The manuscript submitted by Wen et al. isolated and characterized the CUC2 gene from Liriodendron chinense. The authors showed that LcCUC2 promoter directs the expression of a reporter to cotyledons and roots in Arabidopsis. In line with the expression pattern, overexpression of LcCUC2 strongly affects cotyledon shape and vascular patterning, and the number and size of real leaves. Finally, they showed that the phenotypes observed in plants overexpressing LcCUC2 might be due to the differential regulation of auxin content and genes involved in auxin signaling and leaf shape.
The data presented in this MS contribute to the understanding of CUC2 role in leaf development in L. chinense. Overall the MS is well-written, and accompanied with high quality experiments. This manuscript would seem as a good candidate for publication in PeerJ after adequate revision of few issues outlined below.

Major concerns:
- The promoter region of LcCUC2 presents several auxin responsive elements. Is LcCUC2 expression regulated by auxin? Is there any feedback loop as shown for CUC2 and auxin in Arabodopsis?
- The analysis of plants overexpressing LcCUC2 seems to be done using one transgenic line. Do the authors see a similar phenotype for other lines? At least a second line should be included.
- Liriodendron chinense presents one deep lobe on each side of the leaf margin and CUC2 transcript levels are regulated by miR164 to control the extension of leaf serration. Can be LcCUC2 also recognized by miR164? How can be explained that leaves do not seem to have enhanced serration in LcCUC2 overexpression plants?
- In Fig. 3B, demonstration of vascular patterning should be improved. This could be done by using 5(6)-carboxyfluorescein diacetate (CFDA).

Other concerns:
- Line 56: Liriodendron. chinense should be Liriodendron chinense (without the dot).
- Line 95: Since all described phenotypes are from the literature, I would suggest to rephrase to “Taken together, these observations suggest that CUC2 participates in both growth and development in plants”.
- Line 198: Please state the organisms used to align the amino acid sequence of LcCUC2.
- Fig. 2A: If possible, change the color of the graph (Red/Green) as they cannot be distinguished by colorblind readers.
- Line 245: Meaning of SEM should be included.
- Line 258: The statement “expression of some genes” is too vague. Please rephrase.

Reviewer 2 ·

Basic reporting

please see general comments for authors.

Experimental design

please see general comments for authors.

Validity of the findings

please see general comments for authors.

Additional comments

Wen et al. report the identification of CUC2 gene from Liriodendron chinense and investigation of its possible function using transgenic Arabidopsis plants. The LcCUC2 sequence was analyzed using some databases. Its promoter region was introduced in Arabidopsis plants for tissue expression study and the coding sequence for possible functional study, respectively.

Major comments
Figure1; Some CUC genes contain the target sequence of miR164 (line 83). Does LcCUC2 possess such sequence? Please indicate the color codes in the alignment.

Figure 2A; It must be better to present a bar graph with the statistical analysis of three biological replicates, rather than the heat map. The GUS-positive pattern of LcCUC2 was quite different from that of AtCUC2 (Nikovics et al. Line 482). Please carefully discuss a mechanism of the two different patterns. Please present number of transgenic lines analyzed (see also Figure 3).

Figure 3; Because a transgene is randomly introduced in a plant genome, more than two transgenic lines for 35S:LcCUC2 are preferred in the functional analysis. How many transgenic lines the authors obtained in total? Were they showing similar phenotype? It could be better to determine the transcripts of LcCUC2 in the established transgenic lines. The 35S:AtCUC1 (a homologue of CUC2) plant shows serrated cotyledons and leaves with some adventitious shoots in addition to complicated patterns of vasculature. Differentiation of its epidermal cells appear to be suppressed (Hibara, K., Takada, S., and Tasaka, M. (2003). CUC1 gene activates the expression of SAM-related genes to induce adventitious shoot formation. Plant J. 36: 687–696.) The authors can refer this article and must carefully describe the differences from two over-expressor plants of LcCUC2 and AtCUC1 and discuss a possible mechanism why the two genes act differently.

Fig3C might indicate images of tobacco cells. Preparation of methods, results and figures is confusing.

Figure 4; Please specify the biological numbers of the samples. The RT-PCR data might be cooked using a reference gene before preparation of the graphs.

Other comments
Because L. chinense is not common for many readers, it could be helpful if the authors present photographs of the leaf samples mentioned in this study.

Line 163; the templates might be genome DNA or total RNA being reverse-transcribed.

Line 184; Please specify biological or technical replicates.

Line 189; LcCUC2 gDNA sequence might be found in some databases. Because CUC genes shape a gene family, the reviewer would like to know how LcCUC2 were identified. Does the L. chinense genome have CUC1 and CUC3?

---

## Round 0.2 · Major Revisions

Thank you for making the requested changes to the manuscript. In addition to the comments made by the reviewers please consider these additional comments.

Your analysis includes experiments with sometimes two biological replicates or only technical replicates. Please ensure that you present data for three biological replicates - at a minimum.

Please make a point of emphasizing that LcCUC2-like is functionally different from AtCUC2.

In figure 2E, why is expression of LcCUC2-like not present in all the nuclei? How representative is the figure (i.e. how many replicates)? Why is there DAPI staining in the control? Are the emission channels narrow enough?

In figure 3, different lines show difference in expression of LcCUC2-like. Are the phenotypes of these lines correlated with the different expression levels?

Minor points:

Line 1: should be involved not involving.
Line 2: Chineses should not be capitalized.
Line 86: italicize cuc1 cuc2
Line 104: italicize cuc1 cuc2
Line 180: What cells are you referring to? Agrobacterium? Which strain did you use?
Line 201: Do you mean critical point drying or something else? Please specify.
Line 204: Which rosette leaves did you sample?
Line 308: After screening the... should be - After screening on...
Line 315: change ...they had narrow blades than the WT... to -...they had narrow blades compared to the WT..
Line 412: italicize mutant names.

For the supplementary figures/tables please include legends. I could not locate these.

Reviewer 1 ·

Basic reporting

no comment

Experimental design

no comment

Validity of the findings

no comment

Additional comments

The authors have addressed my original concerns and comments, the manuscript is further improved, and I support publication. I have some minor comments that the authors should implement before publication.

Minor changes:
- Regarding miR164, the revised version states that LcCUC2-like posses a conserved V-motif, which is targeted by miR164. However, CUC2-regulation by miR164 occurs at the transcript level, and Fig. 1B show a protein alignment. The authors should show a nucleotide alignment of the nucleotide sequence that might be targeted by miR164, and comment about LcCUC2-like being recognized by miR164 or not.
- Line 66-67: I would suggest to change “Ma proposed” for either “It has been proposed”, “The authors proposed” or “Ma et al. (2018) proposed”. Something similar should be done in line 343 (for Maugarny) and 347 (for Lui).
- Line 241: Please state whether IAA affects LcCUC2 expression negatively (represses) or positively (induces).

Reviewer 2 ·

Basic reporting

The authors present the results using transgenic plants with more than two lines and revise figures. The authors think that LcCUC2-like and AtCUC2 may have different functions, but detailed mechanisms of the difference are still unknown. Some experiments were conducted one or two biological replicates before analyzing data.

Major comments;
The response letter said ”LcCUC2-like and AtCUC2 may have different functions.” Please clearly indicate it in the main text and abstract.

Do the authors think one deep lobe (line 62) is regulated by LcCUC2-like? Please describe some ideas.

Line 66-68; LcKNOX6 induce leaf lobes and may be affect LcCUC2-like gene. In contrast, it seems likely that leaf margin development is independent on LcCUC2like gene. Please discuss the interaction with the leaf lobe, LcKNOX and LcCUC gene.

Line 307; The transcript level of AtCUC2 increased in LcCUC2-expressing plants compared to the WT A. thaliana plants (Fig. 4E). However, the leaves did not induce serrations. Please present the reason.

Do the authors find some cis-elements that confer differential expression patterns between the promoters of AtCUC2 and LcCUC2 (Table1) ?

Figure2bc legend indicates two biological replicates. Figure 4 presents only technical replicates. At least three biologically independent experiments are usually requested before analyzing data.

Minor comments;
Line 132; A real time PCR device always outputs some cycle numbers. How were the gene levels calculated?
Line 216; Please note that the V motif is composed of amino acids, while the miR164 recognition is based on the nucleotide sequence.
Line 377, AtCUC3 is also involved in the leaf margin development (Hasson et al., 2011).
Table 1; please specify the sequences of the individual motifs.

Experimental design

see above.

Validity of the findings

see above.

Additional comments

see above.

---

## Round 0.3 · Minor Revisions

Thank you for addressing the comments and questions that the reviewers have raised. While I understand that research has been delayed because of the pandemic, it is necessary to have at least three biological replicates in your experiments. If you have 7 lines available, then it should not be an issue.

---

## Round 0.4 · Minor Revisions

Thank you for making the modifications. Before the paper is accepted, please change the title of the paper to better reflect the interpretation of the results. It does not accurately reflect that the gene is involved in leaf development of Liriodendron chinense. Its over expression in a heterologous species does not prove its function in L. chinese. Although it causes cotyledon and rosette phenotypes in Arabidopsis does not show that these are its main functions in L. chinese since there are likely other differences between the CUC2 pathways between Arabidopsis and L. chinese.

---

## Round 0.5 · accepted · Accept

Thank you for making the modifications.